## Registered report

Subject Areas:
psychology

Keywords:
illusory truth effect, awareness, subliminal mere exposure, masking

Author for correspondence:
Kyoshiro Sasaki
e-mail: kyoshiro0920@gmail.com

†The affiliations of the first and third authors have changed since the accepted protocol. The second author joined the study after the protocol was accepted. The second and third authors contributed equally.

# The evasive truth: do mere exposures at the subliminal and supraliminal levels drive the illusory truth effect?

Kyoshiro Sasaki[1], Maiko Kobayashi[2,3,†],
Koyo Nakamura[2,3,4,†] and Katsumi Watanabe[2,5]

[1]Faculty of Informatics, Kansai University, 2-1-1, Ryozenji-cho, Takatsuki, Osaka, 569-1095, Japan
[2]Faculty of Science and Engineering, Waseda University, 3-4-1, Ohkubo, Shinjuku, Tokyo, 169-8555, Japan
[3]Japan Society for the Promotion of Science, Chiyoda-ku, Tokyo, Japan
[4]University of Vienna, Universtätsring 1, 1010 Wien, Austria
[5]University of New South Wales, Sydney, NSW 2052, Australia

 KS, 0000-0002-5496-3748

The subjective truth of a statement is boosted by mere exposure to itself or a part of itself. This phenomenon is referred to as the *illusory truth effect*. We examined whether subliminal pre-exposure to the statement topic would increase its subjective truth. In the exposure phase, participants observed the topic, which was presented supraliminally or subliminally. After the exposure phase, they rated the subjective truth of the statement. If unconscious processing contributed to the illusory truth effect, subliminal exposure to the topic would increase the subjective truth of the statement. On the other hand, if the illusory truth effect required conscious and controlled processing, increases in the subjective truth of a statement would be induced only by supraliminal exposure to the topic. The results showed that the illusory truth effect was not found in either supraliminal or subliminal groups. Our findings provide no reliable evidence that pre-exposure to the statement topic saliently promotes its subjective truth.

## 1. Introduction

'The first author's hobby is photography'. Is this statement true? Presumably, most readers will be unsure of its veracity (though the statement is, in fact, true). How can we increase the subjective truth of such a statement? One way might be to present a statement *repeatedly*, and several anecdotal tips on effective presentation and speaking suggest that important ideas should be repeated to make audiences believe them. In fact, the effectiveness

of repeating a statement has been demonstrated in the field of psychology and has been named the *illusory truth effect* (e.g. [1–4]). These previous studies have demonstrated that statements to which listeners have been exposed are often evaluated to be more truthful.

The illusory truth effect occurs in various contexts. Several studies have found that the mere repeated presentation of a trivial statement increases its subjective truth status (e.g. [2,5,6]). The illusory truth effect also occurs when forming social opinions [5] and in product-related claims (e.g. [7–10]). A meta-analysis of studies in various contexts confirmed that the repetition of statements increases their subjective truth [11]. Thus, the illusory truth effect is robust and has important outcomes in wide-ranging social situations.

One might wonder if the illusory truth effect occurs only when identical statements are repeated. In this regard, Begg *et al*. [12] found that the subjective truth of a statement (e.g. 'The temperature of a hen's body is about 104 degrees Fahrenheit') increased even after pre-exposure to only a part of the statement (e.g. 'A hen's body temperature'). Similarly, Arkes *et al*. [1] found that participants who were previously exposed to arbitrary China-related topics were more likely to judge subsequent China-related statements to be true. In one study, the subjective truth of target claims was found to be boosted by the repetition of associated claims that shared only some words with the target claims [13]. Furthermore, a recent study showed that pre-exposure to statements increased the subjective truth of paraphrases of the same statements [14]. This suggests that the illusory truth effect can be induced by mere exposure to keywords that are semantically related to target statements.

Previous studies have also discussed the underlying mechanism of the illusory truth effect. For example, it was found that repeatedly presented information is judged to be true owing to increased familiarity from mere exposure [3,5]. In particular, Begg *et al*. [3] showed that repetition increases the subjective truth of statements even when the source of the statements is of questionable credibility. The familiarity effect has been hypothesized to derive from processing fluency [15], which is the ease with which perceptual or cognitive information can be processed. In other words, higher processing fluency should correlate with an increase in the subjective truth of statements. For example, rhyming aphorisms, which are processed fluently, are judged to be highly true [16]. Previous studies have identified two possible sources of the illusory truth effect (e.g. [17]), namely the promotion of processing fluency based on repetition (i.e. automatic processing), and the promotion of validity owing to recollection (i.e. controlled processing). These two processes should be independent but interrelated; in general, repetitive exposure to certain stimuli allows for efficient and fluent perceptual processing of the stimuli, frequently leading to the misattribution of perceptual fluency to subjective familiarity [18,19]. In this way, repetition enhances the subjective truth of statements.

Processing fluency can be increased even without conscious exposure to objects and repeated unconscious exposure to objects often evokes positive affective reactions to them—a phenomenon known as the subliminal mere exposure effect (SME effect; e.g. [20,21]). The SME effect is assumed to occur because of familiarity resulting from higher processing fluency [18]; unconscious repeated exposure to objects facilitates cognitive processing of the objects, thereby establishing a preference for them. Given that the illusory truth effect and the SME effect share an underlying cognitive mechanism in terms of mere exposure [11], the familiarity stemming from higher processing fluency plays a key role in both effects. Previous studies on the SME effect have often used a masking paradigm (e.g. [22,23]). In the masking paradigm, a visual mask is used for suppressing the visibility of stimuli presented for an extremely short time (less than 50 ms: e.g. [24]). Preferences for masked stimuli were found to increase despite the participants being unaware of their exposure to these stimuli (e.g. [22,23]). Therefore, it is possible that even SME to parts of certain statements can trigger the illusory truth effect; however, to our best knowledge, there has been no direct evidence for this yet.

The present study examined whether subliminal pre-exposure to parts of statements (i.e. topics) would increase the statements' subjective truth by using the visual masking paradigm. Such an examination may help identify the stages of mental processing involved in the illusory truth effect. As discussed above, both automatic and controlled processing is likely to involve the illusory truth effect (e.g. [17]). If automatic processing was sufficient for the occurrence of the illusory truth effect, subliminal pre-exposure to the topic would induce the illusory truth effect. On the other hand, if conscious access to the topic at the pre-exposure phase (i.e. controlled processing) was necessary, then increases in the subjective truth of statements would be induced only by supraliminal exposure to the topic.

## 2. Method

### 2.1. Participants

We planned to perform a two-way mixed-design analysis of variance (ANOVA), with repetition (repeated versus new) as a within-participant factor and exposure groups (supraliminal versus subliminal) as a

between-participant factor. The required sample sizes were calculated using PANGEA (Westfall, 2016 *PANGEA: power analysis for general ANOVA designs*. Unpublished manuscript: see http://jakewestfall.org/publications/pangea.pdf). Although we set a high-power level (i.e. $1 - \beta = 0.95$), the actual power level is possibly low in designs including several different tests [25]. Generally, the two-way mixed-design ANOVA addresses the tests up to seven effects (i.e. two main effects, an interaction effect, and four simple main effects). Considering this, we decided to adopt a higher power level $0.95^{1/7} = 0.993$ to maintain the high-power level. We performed a preliminary test to detect the main effect of repetition (Cohen's $d = 0.45$, replications = 20, var [error] = 0.333, var [participant $^*$ repetition] = 0.167, $1 - \beta = 0.993$), and the sample size was estimated to be 36 per group (e.g. 72 in total) 0.[1]

Moreover, we conducted a preliminary experiment to confirm whether recollection performance would be different between the exposure methods. In the preliminary experiment, we planned to perform a two-tailed *t*-test for two exposure groups. We calculated the required sample size with G$^*$power ([26]; Cohen's $d = 0.45$, $\alpha = 0.05$, $1 - \beta = 0.95$, allocation ratio = 1). As a result, we considered the sample size $n = 130$ per group (i.e. 260 in total) to be valid in the preliminary experiment.

After the in-principle acceptance, we started data collection. Although we were supposed to collect the sample size as planned in the preliminary experiment, the participant count was exceeded by 1 in the main experiment because the experimenters miscounted the number of participants. As a result, the total sample size in the visible group was 37. We used the data for statistical analyses in the order of the earliest timestamp and also reported the results for all the participants in a footnote (for details see the Results section).

## 2.2. Apparatus

The stimuli were presented on a 23.5-inch LED monitor (FORIS FG2421, EIZO) with a refresh rate of 100 Hz and a screen resolution of $1920 \times 1080$ pixels. Stimuli presentation and data collection were controlled using MATLAB running on a MacMini. The stimuli were generated by MATLAB with the Psychtoolbox extension [27,28].

## 2.3. Stimuli

We selected 60 pairs of true/false statements from Unkelbach & Rom [6] as the stimuli (e.g. 'The most toxic jellyfish in the world is *Chironex fleckeri*' versus 'The most toxic jellyfish in the world is *Cotylorhiza tuberculata*'). The statements were translated into Japanese by a translation agency (Editage: https://www.editage.jp/). Next, we recruited two persons who were unaware of our purpose and asked them to choose a topic from each pair of statements. First, we excluded the pairs of statements from which they picked different words as the topic. Second, we extracted only nouns or proper nouns as topics from the remaining selections. Finally, we excluded the pairs of statements whose topic words had more than 10 characters. Ultimately, 40 pairs of statements and their topics remain to be used as stimuli (please see the electronic supplementary material).

## 2.4. Procedure

The experiment consisted of two phases: exposure and test. In the exposure phase, the participants performed a visibility rating task (figure 1). Participants initiated each trial by pressing the space key, after which a black fixation circle with a 1 deg diameter appeared (i.e. blank screen) for 500 ms. In the supraliminal group, a visual mask (i.e. 'XXXXXXXXXX')[2] appeared for 100 ms, after which the blank screen was presented again for 100 ms. Then, the stimulus (i.e. topic) appeared on the computer

---

[1]One might argue that a preliminary test should be performed to detect the interaction effect. Indeed, we are interested in both the interaction and main effects of repetition. We did perform a preliminary test to detect the interaction effect (Cohen's $d = 0.45$, replications = 20, var [error] = 0.5, var [participant $^*$ repetition] = 0.167, $1 - \beta = .993$), resulting in a required sample size of 36 per group (i.e. 72 in total). Thus, the number of the required sample size was identical to those in the case of detecting the main effect of repetition.

[2]In the accepted protocol, we planned to present the stimuli for 50 ms and the string of 'XXXX' as the visual mask. Although we prepared the programme in line with this, according to our observations, the visibility of the stimuli was too high even in the subliminal condition. Thus, we decided to shorten the presentation duration of the target (i.e. 30 ms) and enlarge the length of the visual mask to completely cover the target with the maximum number of characters in our stimulus set (i.e. 10 characters, 'XXXXX XXXXX'). Accordingly, we also modified figure 1. We reported these minor deviations to the action editor of the previous journal and obtained their approval.

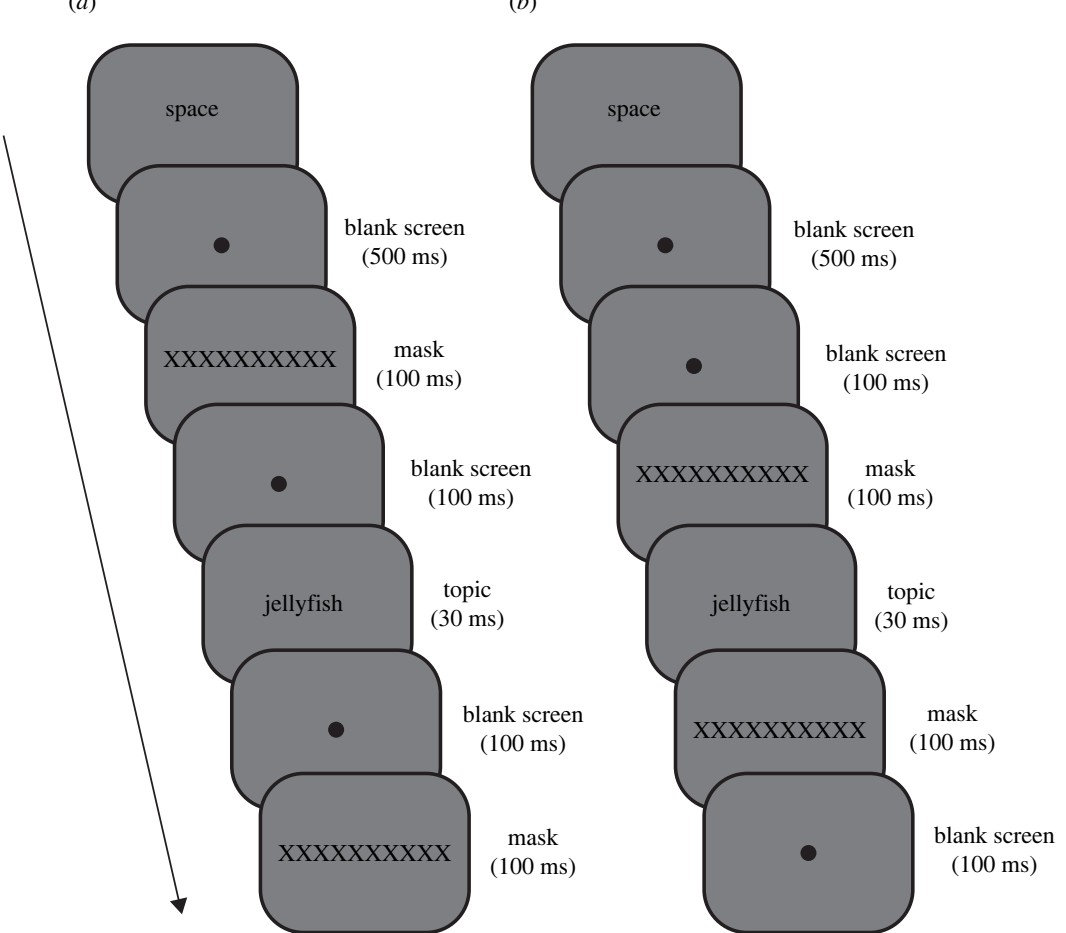

**Figure 1.** An example of the timeline of a single trial in the (*a*) supraliminal and (*b*) subliminal exposure phases.

display for 30 ms.[2] After the topic disappeared, the blank screen was presented again for 100 ms, followed by presentation of the mask for 100 ms. In the subliminal group, the stimulus presentation was identical to the supraliminal condition, except that the temporal order of the mask and blank screen after the initial blank screen was reversed. After stimulus presentation, participants were asked to rate the visibility of the topic on a four-point scale (1: no experience, 2: brief glimpse, 3: almost clear experience, 4: clear experience) derived from a previous study [29]. Prior to the experiment, we provided a detailed explanation of each point of the visibility rating (i.e. no experience = 'no impression of the stimulus. All answers are considered to be mere guesses'; brief glimpse = 'a feeling that something has been shown. Not characterized by any content, and this cannot be specified any further'; almost clear experience = 'ambiguous experience of the stimulus. Some stimulus aspects are experienced more vividly than others. A feeling of almost being certain about one's answer'; clear experience = 'non-ambiguous experience of the stimulus. No doubt in one's answer'). These explanations were taken verbatim from Ramsøy & Overgaard [29]. Before the exposure phase, we told the participants that some words, which had under 10 characters, were presented as stimuli. Twenty topics were randomly selected across the participants. Each topic was presented 10 times, and thus participants performed 200 trials in total. The trial order was randomized across the participants.

After completing the exposure phase, participants performed the test phase. The participants initiated each trial by pressing the space key, after which a statement was presented. They were asked to rate the subjective truth of the statement on a seven-point scale (1 = certain it is false; 7 = certain it is true). We randomly selected one statement from each pair and 40 statements were presented in total; half the statements were true, and the other half were false. Each of the repetition conditions (repeated versus new) included 20 true and 20 false versions. The trial order was randomized across participants. Prior to the test phase, the participants were informed that half the statements are true and the other half false.

To confirm whether recognition of the topics was promoted only by the supraliminal pre-exposure, we also planned to run a preliminary experiment for a different group of participants. We conducted a

recognition phase instead of the rating phase in the preliminary experiment. In the recognition phase, we presented all the topics one-by-one and asked the participants to judge whether each topic was presented in the exposure phase. The other procedure was identical to the main experiment.

## 2.5. Data analysis

We set the α-level to 0.05 and reported effect sizes ($\eta_p^2$, Cohen's $d$ and Cohen's $dz$) for all the analyses. We computed the average scores of the subjective truth for each condition in the main experiment. Then, we performed a two-way mixed-design ANOVA on the subjective truth, with the repetition (repeated versus new) as a within-participant factor and exposure groups (supraliminal versus subliminal) as a between-participant factor. If the interaction effect was statistically significant, we would perform subsequent simple main effect tests. If subliminal pre-exposure to parts of statements (i.e. topics) increased the subjective truth of the statements, the following two cases were expected. One was that the main effect of the repetition would be significant. However, when there was a difference in the effect size of the repetition, the interaction effect would be significant; in this case, the simple main effect of the repetition would be significant in both of the repetition conditions. On the other hand, if the occurrence of the subjective truth effect required conscious access to the topic at the pre-exposure phase, the significant interaction effect would be found and the simple main effect of the repetition would be significant only in the supraliminal condition. To ascertain that the masking procedure rendered the stimuli invisible in the subliminal group, we would perform a two-tailed $t$-test on the visibility scores between the supraliminal and subliminal groups; the scores would be significantly lower in the subliminal group than in the supraliminal group.

Moreover, we computed $d'$ based on the signal detection theory in the preliminary experiment[3] and planned to perform a two-tailed $t$-test on $d'$ between the supraliminal and subliminal groups. If recollection was promoted only by the supraliminal pre-exposure, $d'$ would be higher in the supraliminal group than in the subliminal group.

# 3. Results

All the data are available at the Open Science Framework (OSF) page (https://osf.io/wvru5/).

## 3.1. Preliminary experiment

### 3.1.1. Visibility task

Figure 2a shows the results of the visibility score in the preliminary experiment. The $t$-test revealed that the score was significantly higher in the supraliminal group than in the subliminal group ($t_{258} = 21.589$, $p < 0.001$, Cohen's $d = 2.678$).

### 3.1.2. Recognition task

Figure 2b shows the results of $d'$ in the preliminary experiment. The $t$-test revealed that $d'$ was significantly higher in the supraliminal group than in the subliminal group ($t_{258} = 9.789$, $p < 0.001$, Cohen's $d = 1.214$).

## 3.2. Main experiment[4]

### 3.2.1. Visibility task

Figure 3a shows the results of the visibility score in the main experiment. The $t$-test revealed that the score was significantly higher in the supraliminal group than in the subliminal group ($t_{70} = 12.608$, $p < 0.001$, Cohen's $d = 2.972$).

---

[3]When a hit rate and false alarm rate are 0 or 1.0, $d'$ cannot be calculated. To avoid this, when their rates are under 0.1 and over 0.9, we adjusted them to 0.1 and 0.9, respectively [30]. We missed explaining these in the accepted protocol but determined this method of calculation before checking the collected data.

[4]Here, we report the results for all participants. For the visibility task, the $t$-test revealed that the score was significantly higher in the supraliminal group than in the subliminal group ($t_{71} = 12.812$, $p < 001$, Cohen's $d = 2.999$). For the truth-rating task, the ANOVA

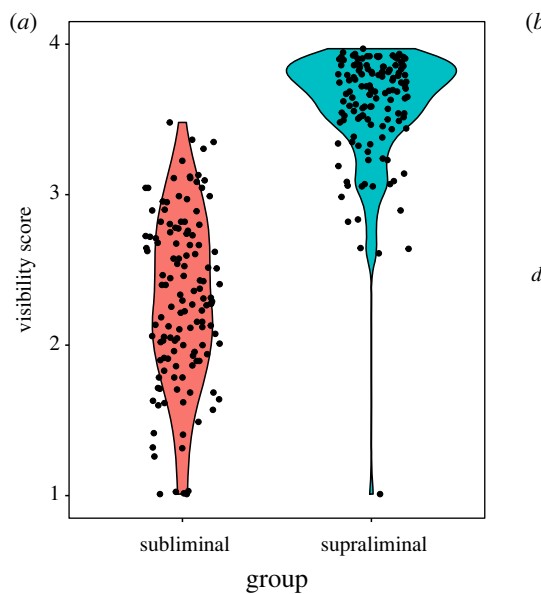
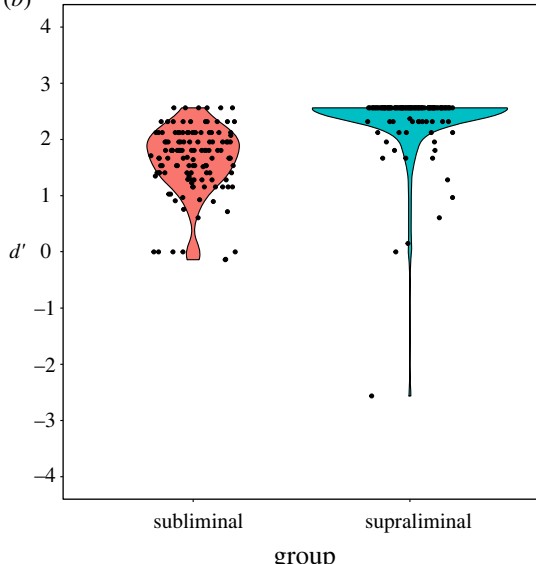

**Figure 2.** The results of the visibility task (*a*) and recognition task (*b*) in the preliminary experiment.

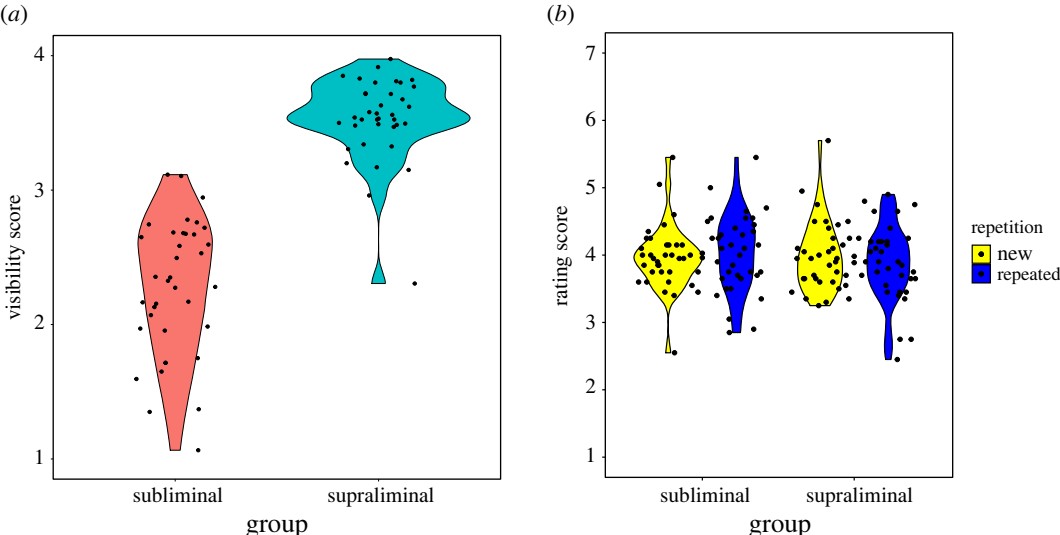

**Figure 3.** The results of the visibility task (*a*) and truth rating task (*b*) in the main experiment.

### 3.2.2. Truth rating task

Figure 3*b* shows the truth rating scores in the main experiment. The ANOVA revealed that the main effects of repetition ($F_{1, 70} = 0.062$, $p = 0.805$, $\eta_p^2 = 0.001$) and exposure groups ($F_{1, 70} = 0.282$, $p = 0.597$, $\eta_p^2 = 0.004$) were not significant. The interaction effect was also not significant ($F_{1, 70} = 2.075$, $p = 0.152$, $\eta_p^2 = 0.029$).

## 4. Discussion

We aimed to examine whether the subjective truth of a statement would increase by subliminal and supraliminal pre-exposure to a part of it. The main effect of repetition and the interaction effect were not significant. Our findings provide no reliable evidence that pre-exposure to a part of the statement has a salient impact on its subjective truth.

revealed that the main effects of repetition ($F_{1, 71} = 0.047$, $p = 0.829$, $\eta_p^2 = 0.001$) and grouping ($F_{1, 71} = 0.242$, $p = 0.624$, $\eta_p^2 = 0.003$) were not significant. Moreover, the interaction effect was not significant ($F_{1, 71} = 2.029$, $p = 0.159$, $\eta_p^2 = 0.028$). The trends in these results are similar to those reported in the main text.

Previous studies have demonstrated that pre-exposure to a trivial statement boosts its subjective truth (e.g. [1–4]). This illusory truth effect was found in the case of pre-exposure to a part of the statement [12]. The supraliminal condition in the current study is a conceptual replication of this; we examined whether pre-exposure to the topic of a statement would increase the subjective truth of the statement. We found no significant difference in the truth scores between the new and repeated conditions in the supraliminal condition. Therefore, we failed to conceptually replicate the results of the previous study [12].

Furthermore, the subliminal group did not present the illusory truth effect. The illusory truth effect is not solely based on the controlled processing but also the automatic processing owing to processing fluency (e.g. [17]). As SME to objects is assumed to evoke processing fluency (e.g. [20,21]), subliminal pre-exposure to the topic of a statement would increase the subjective truth of the statement; however, our results did not support this. Moreover, the present study did not find the illusory truth effect in the supraliminal group; hence, we cannot conclude that the illusory truth effect requires conscious exposure.

The promotion of validity through recollection is assumed to play a key role in the illusory truth effect (e.g. [17,31]). One might argue that recollection is difficult in the set-up of the current study, which, in turn, attenuates the illusory truth effect. However, the $d$'s were not low in the preliminary experiment (subliminal group: $M = 1.643$; supraliminal condition: $M = 2.369$) and the effect sizes compared with 1 were also large (subliminal group: Cohen's $d_z = 1.093$; supraliminal condition: Cohen's $d_z = 2.251$). Thus, recollection performances were unlikely to be low or involved in the failure to replicate the illusory truth effect. Moreover, considering that the $d$'s were larger in the supraliminal than in the subliminal group, the effects of repetition on subjective truth were not significant in either the subliminal or supraliminal groups, and the illusory truth effect was independent of the degrees of recollection. Taken together, it is less likely that recollection was involved in the illusory truth effect, at least in the current study.

Why then were we not able to obtain reliable evidence of the illusory truth effect not salient in the current study? Perhaps, the presentation duration of the topics (i.e. 30 ms) was so short in the exposure phase that they were not invisible even in the supraliminal group. However, the visibility score was not low in the supraliminal group (preliminary experiment: $M = 3.586$; main experiment: $M = 3.546$), and therefore the topics were likely to be visible in the exposure phase. Moreover, although most studies adopted a small number of repetitions (cf. [32]), the current study repeated each topic 10 times during the exposure phase. Considering that a large number of repetitions makes the illusory truth effect more salient (e.g. [13,33–35]), it is unlikely that pre-exposure to the topics was insufficient to produce the illusory truth effect.

Is the illusory truth effect truly robust in the first place? A meta-analysis of studies has shown that repetition affects subjective truth in various contexts [11]. It has also been reported that the illusory truth effect was insusceptible to individual differences in cognitive ability, need for cognitive closure and cognitive style [36]. These studies confirm the robustness of the illusory truth effect. However, a recent study raised concerns about a generalization of the illusory truth effect [37]. It pointed out that the location of research on the illusory truth effect is biased towards Western universities. Indeed, to the best of our knowledge, there are few studies on the illusory truth effect on Japanese people. Therefore, it is possible that the illusory truth effect is not salient among Japanese people. To shed light on the extent to which the illusory truth effect is generalizable, multi-laboratory replications of the illusory truth effect would be warranted.

Visibility scores in the subliminal group were somewhat variable and overlapped with those in the supraliminal group, suggesting our stimulus presentation method may not have been ideal for examining subliminal or non-conscious processing. While this did not bias our current findings—as our findings provide no reliable evidence that pre-exposure to a part of the statement has a noticeable impact on its perceived truth—it does highlight a potential issue. Had we found a significant effect in both subliminal and supraliminal groups, we would not have been able to assert that the illusory truth effect is triggered by subliminal pre-exposure. These potential issues would be avoided only by thoroughly refining the protocol at the review process of the 1st Stage in the Registered Reports system. That is, before the protocol was accepted, we could have conducted preliminary experiments to better mask stimuli or defined a clear visibility score threshold for invisibility, devising an outcome-neutral test that only includes data meeting this criterion for the main hypothesis test. Future studies will address these issues.

**Ethics.** The ethics committee of Waseda University (2015-033) and Kansai University (2020-001) approved the protocol for the present study. All procedures were conducted according to the guidelines contained in the Helsinki Declaration. Written informed consent was obtained from all participants in advance.

**Data accessibility.** Our data, digital materials/code and the approved Stage 1 protocol are available at the OSF page (data and digital materials/code: https://osf.io/wvru5/; protocol: https://osf.io/9wfh8). Data are also provided in the electronic supplementary material [39].

**Authors' contributions.** K.S.: conceptualization, data curation, formal analysis, funding acquisition, investigation, methodology, project administration, resources, software, visualization, writing—original draft, writing—review and editing; M.K.: data curation, formal analysis, funding acquisition, investigation, methodology, resources, software, visualization, writing—review and editing; K.N.: conceptualization, data curation, formal analysis, funding acquisition, methodology, resources, software, visualization, writing—review and editing; K.W.: funding acquisition, resources, supervision, writing—review and editing.

All authors gave final approval for publication and agreed to be held accountable for the work performed therein.

**Conflict of interest declaration.** We declare we have no competing interests.

**Funding.** This research is supported by JSPS KAKENHI (grant nos. JP17J05236, JP19K14482, JP22K13881, JP23H03702 to K.S., JP19K20591 and JP21J01731 to M.K., JP17J04125 to K.N., JP17H06344 and JP22H00090 to K.W., and JP21K18534 and JP23H01046 to K.S. and M.K.), and JST-Moonshot R&D grant (JPMJMS2012) to K.W.

**Acknowledgements.** We hired experimenters who were unaware of our purpose for collecting the data. We would like to thank them. We are deeply grateful to Dr Chris Chambers. The protocol was initially accepted in principle by another journal. However, begrudging withdrawal of the protocol owing to difficulty in meeting the deadline occurred owing to the COVID-19 pandemic (for details, see [38]). At this time, Dr Chambers was instrumental in transferring the in-principle acceptance of our protocol to the *Royal Society Open Science*.

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
