## [Peer Review File · Royal Society Open Science]

Note: This manuscript was transferred from another publisher with peer review (see Appendices A & B).

Review History

Decision letter (RSOS-201791.R1)

Dear Dr Sasaki

On behalf of the Editor, I am pleased to inform you that your Manuscript RSOS-201791 entitled "(Provisional title) Truth out of our awareness: Subliminal mere exposure drives illusory truth effect" has been accepted in principle for publication in Royal Society Open Science. The reviewers' and editors' comments are included at the end of this email.

You may now progress to Stage 2 and complete the study as approved. Before commencing data collection we ask that you:

- 1) Update the journal office as to the anticipated completion date of your study.
- 2) Register your approved protocol on the Open Science Framework (<https://osf.io/>) or other recognised repository, either publicly or privately under embargo until submission of the Stage 2

manuscript. Please note that a time-stamped, independent registration of the protocol is mandatory under journal policy, and manuscripts that do not conform to this requirement cannot be considered at Stage 2. The protocol should be registered unchanged from its current approved state, with the time-stamp preceding implementation of the approved study design.

Following completion of your study, we invite you to resubmit your paper for peer review as a Stage 2 Registered Report. Please note that your manuscript can still be rejected for publication at Stage 2 if the Editors consider any of the following conditions to be met:

- The results were unable to test the authors' proposed hypotheses by failing to meet the approved outcome-neutral criteria.
- The authors altered the Introduction, rationale, or hypotheses, as approved in the Stage 1 submission.
- The authors failed to adhere closely to the registered experimental procedures. Please note that any deviations from the approved experimental procedures must be communicated to the editor immediately for approval, and prior to the completion of data collection. Failure to do so can result in revocation of in-principle acceptance and rejection at Stage 2 (see complete guidelines for further information).
- Any post-hoc (unregistered) analyses were either unjustified, insufficiently caveated, or overly dominant in shaping the authors' conclusions.
- The authors' conclusions were not justified given the data obtained.

We encourage you to read the complete guidelines for authors concerning Stage 2 submissions at <https://royalsocietypublishing.org/rsos/registered-reports#ReviewerGuideRegRep>. Please especially note the requirements for data sharing, reporting the URL of the independently registered protocol, and that withdrawing your manuscript will result in publication of a Withdrawn Registration.

Please note that Royal Society Open Science will introduce article processing charges for all new submissions received from 1 January 2018. Registered Reports submitted and accepted after this date will ONLY be subject to a charge if they subsequently progress to and are accepted as Stage 2 Registered Reports. If your manuscript is submitted and accepted for publication after 1 January 2018 (i.e. as a full Stage 2 Registered Report), you will be asked to pay the article processing charge, unless you request a waiver and this is approved by Royal Society Publishing. You can find out more about the charges at <https://royalsocietypublishing.org/rsos/charges>. Should you have any queries, please contact openscience@royalsociety.org.

Once again, thank you for submitting your manuscript to Royal Society Open Science and we look forward to receiving your Stage 2 submission. If you have any questions at all, please do not hesitate to get in touch. We look forward to hearing from you shortly with the anticipated submission date for your stage two manuscript.

on behalf of Professor Chris Chambers (Registered Reports Editor, Royal Society Open Science)
openscience@royalsociety.org

Editor Comments to Author (Professor Chris Chambers):

Following editorial evaluation of the reviews from *Consciousness and Cognition*, the submitted manuscript (with the noted minor deviations) is hereby granted Stage 1 in-principle acceptance at Royal Society Open Science. No specific deadline is set for submission at Stage 2 but please

update the journal office with an estimated resubmission date and keep the office (or editor) informed of any additional delays.

Author's Response to Decision Letter for (RSOS-201791.R0)

See Appendix C.

RSOS-201791.R1 (Revision)

Review form: Reviewer 1

Is the manuscript scientifically sound in its present form?

Yes

Are the interpretations and conclusions justified by the results?

Yes

Is the language acceptable?

Yes

Do you have any ethical concerns with this paper?

No

Have you any concerns about statistical analyses in this paper?

Yes

Recommendation?

Accept with minor revision

Comments to the Author(s)

Review of Ms. RSOS-201791.R1, "The evasive truth: Do mere exposures at the subliminal and supraliminal levels drive the illusory truth effect?", by Kyoshiro Sasaki, Maiko Kobayashi, Koyo Nakamura, & Katsumi Watanabe

The authors study the so-called "illusory truth effect", where priming with a concept related to the topic of the question increases the truth rating for that sentence. In a large study, they test this effect under conditions of strong or weak masking. Even though there are clear differences in sensitivity and visibility ratings between those two conditions, visibility varies widely between participants, and on average the stimuli are far from invisible, which would have rendered the study inconclusive. The only interesting feature is the total absence of the illusory truth effect in either condition.

I reviewed the preregistration of this article for a previous journal. Because the study has been preregistered, it is probably a sure thing that RSOS will publish it, and I don't see any objection that would preclude publication on technical grounds. However, I have to say that this is a case where preregistration allows a relatively weak dataset to be published that would otherwise not have seen the light of day. In the present case, there is no damage because the absence of an "illusory truth effect" is of potential interest to some readers and may even cast doubt on the existence of that effect. However, consider the possibility that the IT effect would have occurred

under both masking conditions. Then the strength of the masking manipulation would have been insufficient to argue for an unconscious effect, the journal would have felt compelled to publish that inconclusive result, and I the reviewer would have been reluctant to criticize the method because I had approved it previously (but of course under the assumption that the authors would try for more effective masking). In sum, a weak study would have been published ONLY because it had been preregistered. I think that this is a case in point why preregistration may not necessarily lead to higher quality in published results. In any case, it should be considered whether the final paper should be reviewed by the same people who already reviewed the preregistration, because the resulting review process may be pretty lenient – especially if reviews are public.

MAJOR POINTS:

- To argue that the "subliminal" condition is in fact effective, one has to argue that the visibility ratings and the sensitivities are close to zero, not merely that they are lower than in the unmasked case. Overall, the range of values from both measures clearly indicates that the masking was not sufficiently effective. But of course, this point is moot in the absence of indirect effects.

MINOR POINTS:

- The visibility rating categories and category descriptions are verbatim from the Ramsøy and Overgaard paper. This does not become properly clear in the manuscript.
- p13 §2: "...they were not VISIBLE even in the supraliminal group."
- Figures are numbered incorrectly in the text.
- Why does Fig. 3 not show the d' values as Fig. 2 does? The figure should also indicate the scale range [1..7] for the truth scores.

Decision letter (RSOS-201791.R1)

Dear Dr Sasaki:

On behalf of the Editor, I am pleased to inform you that your Stage 2 Registered Report RSOS-201791.R1 entitled "The evasive truth: Do mere exposures at the subliminal and supraliminal levels drive the illusory truth effect?" has been deemed suitable for publication in Royal Society Open Science subject to minor revision in accordance with the referee suggestions. Please find the referees' comments at the end of this email.

The reviewers and Subject Editor have recommended publication, but also suggest some minor revisions to your manuscript. We invite you to respond to the comments and revise your manuscript. Below the referees' and Editors' comments (where applicable) we provide additional requirements. Final acceptance of your manuscript is dependent on these requirements being met. We provide guidance below to help you prepare your revision.

Please submit your revised manuscript and required files (see below) no later than 7 days from today's (ie 15-May-2023) date. Note: the ScholarOne system will 'lock' if submission of the revision is attempted 7 or more days after the deadline. If you do not think you will be able to meet this deadline please contact the editorial office immediately.

Please note article processing charges apply to papers accepted for publication in Royal Society Open Science (<https://royalsocietypublishing.org/rsos/charges>). Charges will also apply to

papers transferred to the journal from other Royal Society Publishing journals, as well as papers submitted as part of our collaboration with the Royal Society of Chemistry (<https://royalsocietypublishing.org/rsos/chemistry>). Fee waivers are available but must be requested when you submit your revision (<https://royalsocietypublishing.org/rsos/waivers>).

on behalf of Professor Chris Chambers
(Registered Reports Editor, Royal Society Open Science)
openscience@royalsociety.org

Associate Editor Comments to Author (Professor Chris Chambers):

Associate Editor: 1

Comments to the Author:

One of the reviewers who evaluated your Stage 1 submission at C&C kindly returned to review the completed Stage 2 manuscript. As you will see, the reviewer has no objection to eventual acceptance of your manuscript. I have also read your article myself and have decided that we can issue an interim decision without requiring additional reviewers.

I want to directly address the following interesting comment by the reviewer, as it bears on the criteria by which Stage 2 RRs are evaluated: "I have to say that this is a case where preregistration allows a relatively weak dataset to be published that would otherwise not have seen the light of day. In the present case, there is no damage because the absence of an "illusory truth effect" is of potential interest to some readers and may even cast doubt on the existence of that effect. However, consider the possibility that the IT effect would have occurred under both masking conditions. Then the strength of the masking manipulation would have been insufficient to argue for an unconscious effect, the journal would have felt compelled to publish that inconclusive result, and I the reviewer would have been reluctant to criticize the method because I had approved it previously (but of course under the assumption that the authors would try for more effective masking). In sum, a weak study would have been published ONLY because it had been preregistered. I think that this is a case in point why preregistration may not necessarily lead to higher quality in published results."

The first point to note is that under RR doctrine, a study is not designated "weak" or "strong" by its data, but by its rationale and design. In this sense, there is arguably a weakness in the *design* in that the confirmation that the subliminal condition is truly targetting subliminal or non-conscious processing is questionable. I fully agree with the reviewer that *had* you found evidence for the illusory truth effect in the subliminal condition, the fact that the visibility scores in the subliminal condition are quite variable (and occasionally overlapping with the superliminal condition) would have raised the question as to whether the masking intervention was sufficiently effective to test the hypothesis. However, if critical, this issue should have been raised during the Stage 1 evaluation at C&C -- for instance, by defining how low the visibility score must be in order to be considered invisible and then devising an outcome-neutral test that e.g. only included data that met this criterion as part of the key hypothesis test. However, the fact is that despite a lengthy set of revisions to improve the design, such a procedure was not considered essential by the C&C editor or reviewers, and so I will not be using as grounds now to reject the Stage 2 submission. This is of course aside from the fact that this potential flaw in the design is moot given the lack of evidence the illusory truth effect in all conditions.

That said, I do believe this issue is important to consider for future studies and replications, so in revising please include a brief consideration in the discussion.

One additional point I noted in my own reading is that the discussion concludes evidence of absence when the statistical procedures employed (conventional NHST) enable only a conclusion of absence of evidence. Specifically on pp11-12 (and potentially elsewhere): "Our findings suggest that pre-exposure to a part of the statement has no salient impact on its subjective truth." Non-significant p values do not permit such a conclusion so please either revise this conclusion (and any others that make the same strong claim) to state e.g. "Our findings provide no reliable evidence that pre-exposure to a part of the statement has a salient impact on its subjective truth" or include exploratory Bayesian hypothesis tests or frequentist equivalence tests in the results to furnish positive evidence of no effect (then keeping the stronger claim). If you decide to conduct such additional tests to permit the current conclusion, then be sure to identify them transparently as unregistered, and if you decide to report Bayesian tests please also report and justify the chosen prior.

Comments to Author:

Reviewer: 1

Comments to the Author(s)

Review of Ms. RSOS-201791.R1, "The evasive truth: Do mere exposures at the subliminal and supraliminal levels drive the illusory truth effect?", by Kyoshiro Sasaki, Maiko Kobayashi, Koyo Nakamura, & Katsumi Watanabe

The authors study the so-called "illusory truth effect", where priming with a concept related to the topic of the question increases the truth rating for that sentence. In a large study, they test this effect under conditions of strong or weak masking. Even though there are clear differences in sensitivity and visibility ratings between those two conditions, visibility varies widely between participants, and on average the stimuli are far from invisible, which would have rendered the study inconclusive. The only interesting feature is the total absence of the illusory truth effect in either condition.

I reviewed the preregistration of this article for a previous journal. Because the study has been preregistered, it is probably a sure thing that RSOS will publish it, and I don't see any objection that would preclude publication on technical grounds. However, I have to say that this is a case where preregistration allows a relatively weak dataset to be published that would otherwise not have seen the light of day. In the present case, there is no damage because the absence of an "illusory truth effect" is of potential interest to some readers and may even cast doubt on the existence of that effect. However, consider the possibility that the IT effect would have occurred under both masking conditions. Then the strength of the masking manipulation would have been insufficient to argue for an unconscious effect, the journal would have felt compelled to publish that inconclusive result, and I the reviewer would have been reluctant to criticize the method because I had approved it previously (but of course under the assumption that the authors would try for more effective masking). In sum, a weak study would have been published ONLY because it had been preregistered. I think that this is a case in point why preregistration may not necessarily lead to higher quality in published results. In any case, it should be considered whether the final paper should be reviewed by the same people who already reviewed the preregistration, because the resulting review process may be pretty lenient – especially if reviews are public.

MAJOR POINTS:

- To argue that the "subliminal" condition is in fact effective, one has to argue that the visibility ratings and the sensitivities are close to zero, not merely that they are lower than in the unmasked

case. Overall, the range of values from both measures clearly indicates that the masking was not sufficiently effective. But of course, this point is moot in the absence of indirect effects.

MINOR POINTS:

- The visibility rating categories and category descriptions are verbatim from the Ramsoy and Overgaard paper. This does not become properly clear in the manuscript.
- p13 §2: "...they were not VISIBLE even in the supraliminal group."
- Figures are numbered incorrectly in the text.
- Why does Fig. 3 not show the d' values as Fig. 2 does? The figure should also indicate the scale range [1..7] for the truth scores.

===PREPARING YOUR MANUSCRIPT===

one version should clearly identify all the changes that have been made (for instance, in coloured highlight, in bold text, or tracked changes);
 a 'clean' version of the new manuscript that incorporates the changes made, but does not highlight them. This version will be used for typesetting.

===PREPARING YOUR REVISION IN SCHOLARONE===

-- If you are requesting an article processing charge waiver, you must select the relevant waiver option (if requesting a discretionary waiver, the form should have been uploaded, see 'File upload' above).

-- If you have uploaded any electronic supplementary (ESM) files, please ensure you follow the guidance at <https://royalsociety.org/journals/authors/author-guidelines/#supplementary-material> to include a suitable title and informative caption. An example of appropriate titling and captioning may be found at https://figshare.com/articles/Table_S2_from_Is_there_a_trade-off_between_peak_performance_and_performance_breadth_across_temperatures_for_aerobic_scope_in_teleost_fishes_/3843624.

Author's Response to Decision Letter for (RSOS-201791.R1)

See Appendix D.

Decision letter (RSOS-201791.R2)

Dear Dr Sasaki:

I am pleased to inform you that your Stage 2 Registered Report entitled "The evasive truth: Do mere exposures at the subliminal and supraliminal levels drive the illusory truth effect?" is now accepted for publication in Royal Society Open Science.

Please remember to make any data sets or code libraries 'live' prior to publication, and update any links as needed when you receive a proof to check - for instance, from a private 'for review' URL to a publicly accessible 'for publication' URL. It is also good practice to add data sets, code and other digital materials to your reference list.

Royal Society Open Science is a fully open access journal. A payment may be due before your article is published. Please note that, if the corresponding author of your paper is based at an institution covered by one of our Transformative Agreement deals, your fees may be covered by the deal – please check the list of eligible institutions at <https://royalsociety.org/journals/authors/read-and-publish/read-publish-agreements/>. The Royal Society has partnered with Copyright Clearance Center's (CCC's) RightsLink service to allow authors to pay article processing charges or page charges. After your manuscript has been accepted, the corresponding author will receive an email from CCC with the subject "Please submit your article processing/open access charge(s)/page charges" inviting you to pay your charges or request an invoice. The email from CCC will come from the email domain @copyright.com (if you have any queries regarding fees, please see <https://royalsocietypublishing.org/rsos/charges> or contact authorfees@royalsociety.org). If you request an invoice, it will be sent to you from CCC. It is important to be cautious about payment scams.

If you receive an email or text message requesting payment and have any concerns, we recommend contacting us through our website, rather than clicking on any links. The Royal Society will never ask you to make a direct payment.

Please see the Royal Society Publishing guidance on how you may share your accepted author manuscript at <https://royalsociety.org/journals/ethics-policies/media-embargo/>. After publication, some additional ways to effectively promote your article can also be found here

<https://royalsociety.org/blog/2020/07/promoting-your-latest-paper-and-tracking-your-results/>.

Your feedback matters - please spend 5 minutes leaving anonymous feedback about your experience of Registered Reports at this journal, as an author or reviewer:

https://registeredreports.cardiff.ac.uk/feedback/feedback/decision_letter.php

This feedback is collected by the Registered Reports Community Feedback website, which is an independent service and research project, being undertaken by Cardiff University.

Kind regards,

on behalf of Professor Professor Chris Chambers (Subject Editor).

<https://www.facebook.com/RoyalSocietyPublishing/>

Appendix A

Revision requested for CONCOG_2019_243

01/Sep/2019 13:54

From: Gregory Francis

To: Kyoshiro Sasaki

BCC:

Ref: CONCOG_2019_243

Title: (Provisional title) Truth out of our awareness: Subliminal mere exposure drives illusory truth effect

Journal: Consciousness and Cognition

Dear Dr. Sasaki,

Thank you for submitting your manuscript to Consciousness and Cognition. The manuscript has been reviewed by two experts, and you can find their thoughts below. I have also read the manuscript myself and added some thoughts below.

Everyone has some enthusiasm for the basic idea of the experiment, but they also have concerns about the proposed design and interpretation. Reviewer 1 has concerns about the methods for establishing subliminal processing, and I think he raises important points. Reviewer 2's concerns are mostly about framing and interpretation. In addition to those concerns, I don't think the presented power calculation actually addresses the planned data analysis, and it is possible your experiment is substantially underpowered.

I am not sure that a revision can address all of these concerns, but perhaps it can be done. Thus, I invite you to resubmit your manuscript after addressing all reviewer comments. I will send the revision back to the reviewers for their expert evaluation.

When resubmitting your manuscript, please carefully consider all issues mentioned in the reviewers' comments, outline every change made point by point, and provide suitable rebuttals for any comments not addressed.

To submit your revised manuscript:

- Log into EVISE® at:
http://www.evise.com/evise/faces/pages/navigation/NavController.jsx?JRNL_ACR=CONCOG
- Locate your manuscript under the header 'My Submissions that need Revisions' on your 'My Author Tasks' view
- Click on 'Agree to Revise'
- Make the required edits
- Click on 'Complete Submission' to approve

What happens next?

After you approve your submission preview you will receive a notification that the submission is complete. To track the status of your paper throughout the editorial process, log in to Evise® at:
http://www.evise.com/evise/faces/pages/navigation/NavController.jsx?JRNL_ACR=CONCOG.

Enrich your article to present your research with maximum impact. This journal supports the following Content Innovations:

Data in Brief (optional)

We invite you to convert your supplementary data (or a part of it) into a Data in Brief article. Data in Brief articles are descriptions of the data and associated metadata which are normally buried in supplementary material. They are actively reviewed, curated, formatted, indexed, given a DOI and freely available to all upon publication. Data in Brief should be uploaded with your revised manuscript directly to Consciousness and Cognition. If your Consciousness and Cognition research article is accepted, your Data in Brief article will automatically be

transferred over to our new, fully Open Access journal, Data in Brief, where it will be editorially reviewed and published as a separate data article upon acceptance. The Open Access fee for Data in Brief is \$500. This fee applies to Data in Brief articles submitted via Consciousness and Cognition between July 1st and December 31st, 2017.

Please just fill in the template found here:

http://www.elsevier.com/inca/publications/misc/dib_data%20article%20template_for%20other%20journals.docx.

Then, place all Data in Brief files (whichever supplementary files you would like to include as well as your completed Data in Brief template) into a .zip file and upload this as a Data in Brief item alongside your Consciousness and Cognition revised manuscript. Note that only this Data in Brief item will be transferred over to Data in Brief, so ensure all of your relevant Data in Brief documents are zipped into a single file. Also, make sure you change references to supplementary material in your Consciousness and Cognition manuscript to reference the Data in Brief article where appropriate.

Questions? Please send your inquiries to dib@elsevier.com. Example Data in Brief can be found here: <http://www.sciencedirect.com/science/journal/23523409>

I look forward to receiving your revised manuscript as soon as possible.

Kind regards,

Dr. Francis
Associate Editor
Consciousness and Cognition

Comments from the editors and reviewers:

-Editor

- I think the authors propose an interesting experiment, but I don't think the document makes a strong case that the experiment is a good one. I am mostly concerned about power of the experiment. I think the current analysis overestimates experimental power. Here are a few specific concerns.

1) It seems it is critical for the supraliminal condition to produce the illusory truth effect. If it does not, then there is nothing else to conclude. Thus, the experiment needs to be designed as such, and the document needs to make a good case that the experiment will show it. To me, that means using established methods and identifying effect sizes for those methods. Then, using that background identifying a sample size that produces very high power (say, 95%).

2) The current power analysis focuses on an interaction, but the hypothesized effect size is unjustified. Moreover, the data analysis section indicates that other tests will also be included, but that can only reduce power, so the sample sizes need to reflect the reduced power that comes with multiple tests. The $n=20$ per condition heuristic that is used seems like a cop out. You can (and should) do better.

-Reviewer 1

-

Review of Report Registration CONCOG_2019_243, provisional title: "Truth out of our awareness: Subliminal mere exposure drives illusory truth effect", by Kyoshiro Sasaki, Koyo Nakamura, and Katsumi Watanabe

Reviewer: Thomas Schmidt

The authors' plan is to test the interesting possibility that mere exposure of factual statements influences the subjective truth of such statements when they are later repeated, compared with novel statements. In the exposure phase, statements are flashed either for 100 ms, followed by a 200-ms mask of Xs, or for a full second without a mask. Subsequently, participants perform a test phase where they are faced with old and new statements and indicate the subjective truth of the items on a rating scale.

Reviewing a registered report requires a different mindset than reviewing a paper. We are probably all aware of the worst-case scenario for registered reports, which is that they become mandatory and degrade into some kind of Orwellian mind police. The challenge for me is to find a balance between providing constructive criticism and forcibly imposing my own methodological utopia on my colleagues, which is not my place as a reviewer. The task is all the more difficult as my standards are quite a bit stricter than those currently followed by most of the field of consciousness science.

MAJOR POINTS

- There will be two unpleasant artifacts in the presentation: Prime and mask take only 300 ms, while the unmasked primes takes a full second; and masked and unmasked primes differ greatly in energy (roughly, $\ln k \times \text{time}$). Both factors would be able to explain any differences between conditions. Another concern is that pattern masking seems to be one of the most disruptive techniques for diminishing visibility and probably interferes with the bottom-up signal of the masked stimulus. Therefore, if the exposure effects do not work under pattern masking (which is what I would expect), no strong conclusions would follow for the possibility of unconscious exposure effect.

- In the exposure phase, participants have a dummy task requiring them to indicate the color of the fixation point. In the group receiving unmasked primes, this is the only task. But the group receiving masked primes has a dual task: they are supposed to press yet another key "when the stimuli are visible" and then to report "what was presented". These are rather unspecific tasks, and it is not clear at all what aspect of the stimulus is supposed to be judged – the presence or absence, the content, or what? It is also not clear how and when the report is performed. Even more problematic: Since every trial contains a stimulus, it is a detection task where every "no" response (every failure to press the key) is incorrect, so that the task is completely determined by the observer's response criterion (bias). What is needed here is a full signal-detection design with stimuli present or absent and responses "present" or "absent", so that sensitivity and bias can be determined independently. But are the authors really interested in the mere presence/absence, or in other aspects of discriminability or subjective appearance? If so, the task has to be chosen accordingly.

- Another practice in this research plan is definitely no longer state of the art: post-hoc exclusion of participants (or trials) whose reports exceed some criterion of visibility. Even though this is common practice, it is not only subject to criterion effects and biases the sample values without taking care of the actual visibility underlying the performance (Schmidt, 2015), it is also subject to regression to the mean and thereby underestimates the degree of conscious visibility (Shanks, 2017). In my opinion, the whole business of post-hoc sorting of trials into different visibility classes (instead of establishing different visibility conditions by varying the stimuli experimentally) is highly problematic and not suited to convince a skeptic.

MINOR POINTS

- Why are conscious and unconscious trials presented in different groups?

- Power in the present experiment will be determined by two terms: the number of participants and the number of trials per participant. Simulation papers have long shown that both layers are about equally important, yet power analysis invariably focuses on the number of participants. The authors use it only to determine their group sizes, but they should also provide some criteria of how exact the measurements will be within each subject and cell of the design. What are the expected standard errors per cell and subject?

-Reviewer 2

-

The idea of testing whether subliminal exposure to the topic of statements can produce illusions of truth is interesting and in principle worth pursuing. However, I believe the framing of the hypotheses and the interpretation of the possible results need to be better framed within the literature addressing the truth effect.

Both in the abstract and in the final paragraph of the introduction, the authors frame their experiment as a test to conclude whether semantic processing contributes to illusions of truth. In the authors' words:

"If unconscious processing contributes to the illusory truth effect, subliminal exposure to the topic will increase the subjective truth of the statements. On the other hand, if conceptual and semantic processing are involved in the illusory truth effect, increases in the subjective truth of statements will be induced only by supraliminal exposure to the topic." (abstract)

As it has been shown in previous studies, illusions of truth due to previous exposure can be driven by different processes, some of which are more automatic and others more controlled, such as increases in processing fluency and recollection, respectively. The paper by Unkelbach and Stahl (2009; A multinomial modeling approach to dissociate different components of the truth effect. *Consciousness and Cognition* 18, 22–38), for example, has addressed and shown this. Thus, if the authors find that indeed subliminal exposure can promote illusions of truth, this does not mean that conceptual and semantic processing do not contribute or are not involved in the truth effect. It merely indicates that under those circumstances (i.e., under subliminal exposure), a rather automatic process is driving the effect.

The authors should also discuss their hypotheses in light of the referential theory of the repetition-based truth effect, put forth by Unkelbach and Rom (2017; A referential theory of the repetition-induced truth effect. *Cognition*, 160, 110–126). This theory proposes that people judge truth based on coherent references for statements in memory, and due to prior presentation, repeated statements have more coherently linked references. Since this is closely linked to semantic processing of information, this referential theory should be taken into consideration both in the rationale for the hypotheses and the discussion of the findings.

As a suggestion, the experiment could include a recognition test after the truth ratings task, in which participants' memory would be probed for the topics (vs. distractors) presented in the exposure phase. The results of this recognition test will bring more insight regarding participant's level of access to the information presented before, allowing for stronger conclusions as to whether some semantic processing occurred in the subliminal exposure condition. The memory test can be made more detailed through the inclusion of synonyms of the words that were presented as topics.

The text of this registered report has some room for improvement in what concerns the use of English. While there are no major language mistakes, the text sounds "weird" at times and this should be improved in the final manuscript.

A final detail, the authors use the acronym SMME to refer to Subliminal Mere Exposure effects; isn't it only SME effects?

MethodsX (optional)

We invite you to submit a method article alongside your research article. This is an opportunity to get full credit for the time and money you have spent on developing research methods, and to increase the visibility and impact of your work.

If your research article is accepted, your method article will be automatically transferred over to the open access journal, MethodsX, where it will be editorially reviewed and published as a separate method article upon acceptance. Both articles will be linked on ScienceDirect.

Please use the MethodsX template available here when preparing your article:
<https://www.elsevier.com/MethodsX-template>. Open access fees apply.

Have questions or need assistance?

For further assistance, please visit our Customer Support site. Here you can search for solutions on a range of topics, find answers to frequently asked questions, and learn more about EVISE® via interactive tutorials. You can also talk 24/5 to our customer support team by phone and 24/7 by live chat and email.

Copyright © 2018 Elsevier B.V. | Privacy Policy

Elsevier B.V., Radarweg 29, 1043 NX Amsterdam, The Netherlands, Reg. No. 33156677.

Appendix B

Revision requested for CONCOG_2019_243_R1

25/Jan/2020 11:24

From: Gregory Francis

To: Kyoshiro Sasaki

BCC:

Ref: CONCOG_2019_243_R1

Title: (Provisional title) Truth out of our awareness: Subliminal mere exposure drives illusory truth effect

Journal: Consciousness and Cognition

Dear Dr. Sasaki,

Thank you for submitting your manuscript to Consciousness and Cognition. The manuscript has been reviewed by the same two experts as the original submission. Both experts agree that the revision substantially improves the proposed work. My impression is similar.

As you can see in the comments below, there remain a few lingering issues that should be addressed. My impression is that these are minor enough that I can evaluate the revised manuscript myself. Thus, I will most likely make a final decision without sending the revision back to the experts.

When resubmitting your manuscript, please carefully consider all issues mentioned in the reviewers' comments, outline every change made point by point, and provide suitable rebuttals for any comments not addressed.

To submit your revised manuscript:

- Log into EVISE® at:
http://www.evise.com/evise/faces/pages/navigation/NavController.jsx?JRNL_ACR=CONCOG
- Locate your manuscript under the header 'My Submissions that need Revisions' on your 'My Author Tasks' view
- Click on 'Agree to Revise'
- Make the required edits
- Click on 'Complete Submission' to approve

What happens next?

After you approve your submission preview you will receive a notification that the submission is complete. To track the status of your paper throughout the editorial process, log in to EVISE® at:
http://www.evise.com/evise/faces/pages/navigation/NavController.jsx?JRNL_ACR=CONCOG.

Enrich your article to present your research with maximum impact. This journal supports the following Content Innovations:

Data in Brief (optional)

We invite you to convert your supplementary data (or a part of it) into a Data in Brief article. Data in Brief articles are descriptions of the data and associated metadata which are normally buried in supplementary material. They are actively reviewed, curated, formatted, indexed, given a DOI and freely available to all upon publication. Data in Brief should be uploaded with your revised manuscript directly to Consciousness and Cognition. If your Consciousness and Cognition research article is accepted, your Data in Brief article will automatically be transferred over to our new, fully Open Access journal, Data in Brief, where it will be editorially reviewed and published as a separate data article upon acceptance. The Open Access fee for Data in Brief is \$500. This fee applies to Data in Brief articles submitted via Consciousness and Cognition between July 1st and December 31st, 2017.

Please just fill in the template found here:

http://www.elsevier.com/inca/publications/misc/dib_data%20article%20template_for%20other%20journals.docx.

Then, place all Data in Brief files (whichever supplementary files you would like to include as well as your completed Data in Brief template) into a .zip file and upload this as a Data in Brief item alongside your Consciousness and Cognition revised manuscript. Note that only this Data in Brief item will be transferred over to Data in Brief, so ensure all of your relevant Data in Brief documents are zipped into a single file. Also, make sure you change references to supplementary material in your Consciousness and Cognition manuscript to reference the Data in Brief article where appropriate.

Questions? Please send your inquiries to dib@elsevier.com. Example Data in Brief can be found here: <http://www.sciencedirect.com/science/journal/23523409>

I look forward to receiving your revised manuscript as soon as possible.

Kind regards,

Dr. Francis
Associate Editor
Consciousness and Cognition

Comments from the editors and reviewers:

-Editor

- The authors did a good job on the revision.

I still have a few lingering concerns about the power analysis and also some concern about the data analysis.

Power: The use of PANGEA is very good, but I worry that the analysis still focuses on just one test (a main effect). Your analysis indicates that you will look for an interaction and main effects and also use a t-test to verify the effectiveness of the mask. OK, but then for your experiment to succeed, you need to have power for all of those tests. Even if the used main effect is the weakest of your proposed tests, the probability of picking a sample that satisfies just that weakest test by itself must overestimate the probability of a picking a sample that satisfies that weakest test and the other relevant tests.

Data analysis: I think the end of the manuscript will benefit by explaining how conclusions will be drawn from the results of the data analysis. For example, suppose you get the interaction but not the main effects? What does that mean? How do you interpret the results if the masking does not seem to work? You don't need to consider every permutation of possible outcomes, but you should at least explain what you think is most favorable for the interpretation you want to make.

Minor:

Introduction, para 1, line 10: delete "previously"

Two lines above "Method" header: maybe "then increases in"

-Reviewer 1

-

Review of Report Registration CONCOG_2019_243_R1, provisional title: "Truth out of our awareness: Subliminal mere exposure drives illusory truth effect", by Kyoshiro Sasaki, Koyo Nakamura, and Katsumi Watanabe

Reviewer: Thomas Schmidt

The authors have substantially revised their research plan. 1) They removed the possible artifacts caused by different timing in visible and invisible conditions. 2) They replaced the rather informal questions about stimulus

visibility with the more formalized Visual Awareness Scale (VAS) and apply it in both the visible and invisible conditions so that there are now identical task requirements in both conditions. 3) They refrain from post-hoc selection of participants or trials. 4) They intermix visible and invisible conditions. 5) They give new power calculations.

I find those changes very convincing and hope that they will help the authors to carry out their project. There is one suggestion that I might add. The VAS scale is currently quite popular, but I have seen several instances where it discriminated rather poorly between perceptual states. For masked stimuli, participants tend to give the rating 2 ("brief glimpse"), but there are extremely low ratings in conditions that should be visible, and also a surprising number of extremely high ratings in catch trials. This poor performance may be the result of applying the scale without further instruction, so that it summarily refers to "the stimulus". In their original paper, Ramsøy and Overgaard instructed participants to look out for specific stimulus features and to apply the PAS rating specifically to those features. This seems to have yielded more discriminatory power.

-Reviewer 2

- I'd like to congratulate the authors on the revision they did to the previous version of the registered report. The response letter and the changes to the report were very easy to follow.

I still have some comments and suggestions that I believe the authors should implement to increase the value of the experiment and the conclusions that can be taken from it. I write them below (the order in which I present my comments/suggestions does not reflect different degrees of importance between them).

The first suggestion pertains to a point in my previous review, the implementation of a recognition test to understand participant's level of access to the information presented before. I do agree with the authors that implementing this measure after the truth ratings task is likely to render it uninformative, given that all participants will then have seen the statements containing the words that are used as topics in the exposure phase. But given that the hypothesis and the experimental procedure are all about one condition allowing conscious perception of the statements' topics and the other not, I suggest the authors implement a recognition test (one allowing the measure of controlled discrimination and of bias in participants' responses) in a separate preliminary experiment, to be able to make more solid claims about the processes underlying their findings.

Another suggestion has to do with the instructions for the truth ratings task, which state the following: "The participants will also be told that the topics of some statements might have been presented in the earlier phase, while those of others might not." Why is this information important? It may actually cause problems for the interpretation of the results. Such an instruction may trigger individuals' beliefs or naïve theories about the ecologically valid connection between repetition and truth (see Silva, Garcia-Marques, & Mello, 2016, for a discussion on this matter) and thus increase the likelihood that participants give higher truth ratings to those statements they can actually remember seeing the topic before. This will in turn reduce the chance of observing effects associated with the subliminal exposure condition, for which there is a lower level of awareness and conscious retrieval.

Regarding the relevance of the referential theory (Unkelbach, Koch, Silva, & Garcia-Marques, 2019; Unkelbach & Rohm, 2017) for the hypothesis being tested in this registered report and the experimental procedure. After going through the materials that will be used in the experiment, it is possible to see to the naked eye how some topics are actually related to each other, as for example the multiple times topics refer to ancient deities or to Italian artists. These topics are related to each other (e.g., all the mentioned gods enter the topic of ancient deities), and so one given statement (e.g., Zeus procreated the three spouses of destiny with Leda) can very well be compatible with the referential network activated by the topic referring to another statement (e.g., Nerthus), because they actually share the same overarching topic (e.g., deities). This may seem like a small detail, but in my opinion it does have an impact for the interpretation of results, especially now that the authors changed the experiment to a within-participants design (regarding this change, the reasons why the authors made this change was not clear in their response to Reviewer 1). For example, let's imagine that in the subliminal exposure condition a participant sees the topic "Bramante" and in the supraliminal exposure the topic "Michelangelo". And then, in the truth ratings phase the participant gives equally higher truth ratings to the statements pertaining to those two topics (as compared to new statements). It is not possible to be entirely sure whether the "similar" evaluations to both repeated statements reflect the fact that subliminal exposure also increases truth-value of information, or if it results from the referential network that is activated by the topic seen on the supraliminal exposure condition.

References

Silva, R. R., Garcia-Marques, T., & Mello, J. (2016). The differential effects of fluency due to repetition and fluency due to color contrast on judgments of truth. *Psychological research*, 80, 821-837.

MethodsX (optional)

We invite you to submit a method article alongside your research article. This is an opportunity to get full credit for the time and money you have spent on developing research methods, and to increase the visibility and impact of your work.

If your research article is accepted, your method article will be automatically transferred over to the open access journal, MethodsX, where it will be editorially reviewed and published as a separate method article upon acceptance. Both articles will be linked on ScienceDirect.

Please use the MethodsX template available here when preparing your article:
<https://www.elsevier.com/MethodsX-template>. Open access fees apply.

Have questions or need assistance?

For further assistance, please visit our Customer Support site. Here you can search for solutions on a range of topics, find answers to frequently asked questions, and learn more about EVISE® via interactive tutorials. You can also talk 24/5 to our customer support team by phone and 24/7 by live chat and email.

Copyright © 2018 Elsevier B.V. | Privacy Policy

Elsevier B.V., Radarweg 29, 1043 NX Amsterdam, The Netherlands, Reg. No. 33156677.

Appendix C

Dr. Christopher Chambers
Editor of *Royal Society Open Science*

4 April 2023

Dear Dr. Chambers,

Please find enclosed our manuscript (ID: RSOS-201791: <https://osf.io/9wfh8/>) for publication as *Registered Reports* in *Royal Society Open Science*. Considering the results of our experiments, we have changed the title from ‘(Provisional title) Truth out of our awareness: Subliminal mere exposure drives illusory truth effect’ to ‘The evasive truth: Do mere exposures at the subliminal and supraliminal levels drive the illusory truth effect?’. The revised and added parts from the protocol manuscript (i.e. the manuscript accepted in principle at the 1st stage) are highlighted in blue.

The subjective truth of statements can be boosted by mere exposure to their topics. This phenomenon is referred to as ‘the *illusory truth effect*’. This effect has been observed in a wide range of social situations, and the underlying mechanism has been vigorously discussed in the field of psychology. However, to the best of our knowledge, it remains unclear whether the subjective truth of the statements can also be boosted even by mere exposure to their topic without conscious awareness.

To address these issues, we prepared a protocol manuscript in which we planned to examine whether subliminal and supraliminal exposures to the topics of the statements would promote the subjective truth of the statements. After the protocol manuscript was accepted, we conducted the experiments and found that neither method of exposure had a significant impact on the subjective truth of the statements. In other words, the illusory truth effect was not salient among Japanese people. The current study provides insights into the extent to which the illusory truth effect is generalisable and will be of interest to the broad readership of *Royal Society Open Science*, as it encompasses the fields of Cognitive Psychology, Consciousness Science, and Social Psychology.

Our protocol manuscript exhibits only minor deviations in the methods from the

accepted version. We have reported these in the Footnotes section. Moreover, after our protocol manuscript was accepted, another member (Maiko Kobayashi) joined and the affiliations of the first (Kyoshiro Sasaki) and third (Koyo Nakamura) authors changed. Our data, digital materials/code, and the approved Stage 1 protocol are available on the OSF page (data and digital materials/code: <https://osf.io/wvru5/> ; protocol: <https://osf.io/9wfh8>). These URL are shown on Page 21 of our manuscript.

This research was supported by JSPS KAKENHI (17J05236, 19K14482, and 22K13881 to K.S., 19K20591 and 21J01731 to M.K., 17J04125 to K.N., 17H06344 and 22H00090 to K.W., and JP21K18534 to K.S. and M.K.), Core Research for Evolutional Science and Technology (MJCR14E4 to K.W.), and a JST-Moonshot R&D Grant (JPMJMS2012 to K.W.). We used the facilities in our laboratory at Kansai University and Waseda University. The ethics committees of Kansai University (2020-001) and Waseda University (2015-033) approved the study protocol. All procedures were conducted in accordance with the guidelines established by the Declaration of Helsinki.

Yours faithfully,

Kyoshiro Sasaki, Ph.D.

Associate Professor

Faculty of Informatics, Kansai University,

2-1-1, Ryozenji-cho, Takatsuki, Osaka, 569-1095, Japan.

E-mail: kyoshiro0920@gmail.com

TEL & FAX: +81-72-690-2447

Appendix D

Dr. Chris Chambers
Registered Reports Editor
Royal Society Open Science

18 May 2023

Dear Dr. Chambers

We have read your and the reviewer's comments regarding our manuscript and we greatly appreciate the valuable feedback. Please find attached the revised manuscript (RSOS-201791.R1) entitled, "The evasive truth: Do mere exposures at the subliminal and supraliminal levels drive the illusory truth effect?" by Sasaki (myself), Kobayashi, Nakamura, and Watanabe. We substantially revised the manuscript in response to your and the reviewer's comments. Below, please find our detailed responses to individual comments.

Responses to Editor

Comments & Replies

E-1 *One of the reviewers who evaluated your Stage 1 submission at C&C kindly returned to review the completed Stage 2 manuscript. As you will see, the reviewer has no objection to eventual acceptance of your manuscript. I have also read your article myself and have decided that we can issue an interim decision without requiring additional reviewers.*

Reply: We cordially thank you for your careful consideration and flexibility in handling our manuscript. The feedback provided has been invaluable in refining our manuscript, and we appreciate the opportunity to improve the manuscript.

E-2 *I want to directly address the following interesting comment by the reviewer, as it bears on the criteria by which Stage 2 RRs are evaluated: "[—Reviewer 1's comment—]"*

*The first point to note is that under RR doctrine, a study is not designated "weak" or "strong" by its data, but by its rationale and design. In this sense, there is arguably a weakness in the *design* in that the confirmation that the subliminal condition is truly targetting subliminal or non-conscious processing is questionable. I fully agree with the reviewer that *had* you found evidence for the illusory truth effect in the subliminal condition, the fact that the visibility scores in the subliminal condition are quite variable (and occasionally overlapping with the superliminal condition) would have raised the question as to whether the masking intervention*

was sufficiently effective to test the hypothesis. However, if critical, this issue should have been raised during the Stage 1 evaluation at C&C -- for instance, by defining how low the visibility score must be in order to be considered invisible and then devising an outcome-neutral test that e.g. only included data that met this criterion as part of the key hypothesis test. However, the fact is that despite a lengthy set of revisions to improve the design, such a procedure was not considered essential by the C&C editor or reviewers, and so I will not be using as grounds now to reject the Stage 2 submission. This is of course aside from the fact that this potential flaw in the design is moot given the lack of evidence the illusory truth effect in all conditions.

That said, I do believe this issue is important to consider for future studies and replications, so in revising please include a brief consideration in the discussion.

Reply: Like you, we concur with Reviewer 1's opinion regarding the weakness in our design. Since our paradigm was based on a previous study (e.g., Cox et al., 2018), we were also surprised to find that our target stimuli were not masked enough. However, as you mentioned, we should have addressed these issues at the Stage 1 process to avoid the potential problem. That is, we believe that the issues raised are not inherent problems of the Registered Reports system, but rather reflect the specific circumstances of our study and its review process. We realized once again that in the Registered Reports system, the experimental design needs to be thoroughly refined to be suitable for hypothesis testing during the 1st Stage process. In the Discussion section of the revised manuscript, we have added the following (p14 lines 313-325): *“Visibility scores in the subliminal group were somewhat variable and overlapped with those in the supraliminal group, suggesting our stimulus presentation method may not have been ideal for examining subliminal or non-conscious processing. While this did not bias our current findings — as our findings provide no reliable evidence that pre-exposure to a part of the statement has a noticeable impact on its perceived truth — it does highlight a potential issue. Had we found a significant effect in both subliminal and supraliminal groups, we would not have been able to assert that the illusory truth effect is triggered by subliminal pre-exposure. These potential issues would be avoided only by thoroughly refining the protocol at the review process of the 1st Stage in the Registered Reports system. That is, before the protocol was accepted, we could have conducted preliminary experiments to better mask stimuli or defined a clear visibility score threshold for invisibility, devising an outcome-neutral test that only includes data meeting this criterion for the main hypothesis test. Future studies will address these issues.”*

E-3 *One additional point I noted in my own reading is that the discussion concludes evidence of absence when the statistical procedures employed (conventional NHST) enable only a conclusion of absence of evidence. Specifically on pp11-12 (and potentially elsewhere): “Our findings suggest that pre-exposure to a part of the statement has no salient impact on its subjective truth.” Non-significant p values do not permit such a conclusion so please either revise this conclusion (and any others that make the same strong claim) to state e.g. “Our findings provide no reliable evidence that pre-exposure to a part of the statement has a salient impact on its subjective truth” or include exploratory Bayesian hypothesis tests or frequentist equivalence tests in the results to furnish positive evidence of no effect (then keeping the stronger claim). If you decide to conduct such additional tests to permit the current conclusion, then be sure to identify them transparently as unregistered, and if you decide to report Bayesian tests please also report and justify the chosen prior.*

Reply: According to your suggestions, we revised some descriptions to weaken our claim (p2 line 37; p12 line 257; p13 line 289). Thanks to your comments, our tone has become more appropriate.

Reference

Cox, E. J., Sperandio, I., Laycock, R., & Chouinard, P. A. (2018). Conscious awareness is required for the perceptual discrimination of threatening animal stimuli: a visual masking and continuous flash suppression study. *Consciousness and Cognition*, 65, 280–292.

Responses to Reviewer 1

Comments & Replies

1-1 *The authors study the so-called "illusory truth effect", where priming with a concept related to the topic of the question increases the truth rating for that sentence. In a large study, they test this effect under conditions of strong or weak masking. Even though there are clear differences in sensitivity and visibility ratings between those two conditions, visibility varies widely between participants, and on average the stimuli are far from invisible, which would have rendered the study inconclusive. The only interesting feature is the total absence of the illusory truth effect in either condition.*

Reply: We would like to thank Reviewer 1 for their understanding of our manuscript.

1-2 *I reviewed the preregistration of this article for a previous journal. Because the study has been preregistered, it is probably a sure thing that RSOS will publish it, and I don't see any objection that would preclude publication on technical grounds. However, I have to say that this is a case where preregistration allows a relatively weak dataset to be published that would otherwise not have seen the light of day. In the present case, there is no damage because the absence of an "illusory truth effect" is of potential interest to some readers and may even cast doubt on the existence of that effect. However, consider the possibility that the IT effect would have occurred under both masking conditions. Then the strength of the masking manipulation would have been insufficient to argue for an unconscious effect, the journal would have felt compelled to publish that inconclusive result, and I the reviewer would have been reluctant to criticize the method because I had approved it previously (but of course under the assumption that the authors would try for more effective masking). In sum, a weak study would have been published ONLY because it had been preregistered. I think that this is a case in point why preregistration may not necessarily lead to higher quality in published results. In any case, it should be considered whether the final paper should be reviewed by the same people who already reviewed the preregistration, because the resulting review process may be pretty lenient – especially if reviews are public.*

Reply: We concur with Reviewer 1's critique regarding the weaknesses in our design. We recognize that these points should have been thoroughly addressed during the 1st Stage review process. After the protocol is accepted, we adhere to the principle of not altering critical aspects of the experiment, thus we could not attempt more effective masking. As such, before the protocol was accepted, we should have performed other preliminary experiments to explore the way of effective masking or clearly defined the threshold for visibility scores to be considered invisible and devised an outcome-neutral test that, for instance, only included data meeting this criterion as part of the key hypothesis test.

However, we would like to respectfully point out that the issues raised are not inherent problems of preregistration, but rather reflect the specific circumstances of our study and its review process. If the protocol had been thoroughly refined to be suitable for testing the main hypothesis during the 1st Stage review process, we could have avoided these issues. We have gained a renewed appreciation for the importance of the 1st Stage review process in Registered Reports.

The above issues are highly important. Therefore, we have included these points in the Discussion section of the revised manuscript as follows (p14 lines 313-325): "*Visibility scores in the subliminal group were somewhat variable and*

overlapped with those in the supraliminal group, suggesting our stimulus presentation method may not have been ideal for examining subliminal or non-conscious processing. While this did not bias our current findings — as our findings provide no reliable evidence that pre-exposure to a part of the statement has a noticeable impact on its perceived truth — it does highlight a potential issue. Had we found a significant effect in both subliminal and supraliminal groups, we would not have been able to assert that the illusory truth effect is triggered by subliminal pre-exposure. These potential issues would be avoided only by thoroughly refining the protocol at the review process of the 1st Stage in the Registered Reports system. That is, before the protocol was accepted, we could have conducted preliminary experiments to better mask stimuli or defined a clear visibility score threshold for invisibility, devising an outcome-neutral test that only includes data meeting this criterion for the main hypothesis test. Future studies will address these issues.”

1-3 *To argue that the “subliminal” condition is in fact effective, one has to argue that the visibility ratings and the sensitivities are close to zero, not merely that they are lower than in the unmasked case. Overall, the range of values from both measures clearly indicates that the masking was not sufficiently effective. But of course, this point is moot in the absence of indirect effects.*

Reply: We would like to thank Reviewer 1 for this comment. Considering this, we added to the description that the masking was less likely to be sufficiently effective in our experiment as we mentioned in the 1-2 reply.

1-4 *The visibility rating categories and category descriptions are verbatim from the Ramsøy and Overgaard paper. This does not become properly clear in the manuscript.*

Reply: In the previous version of our manuscript, we had already cited Ramsøy and Overgaard (2004) when introducing the visibility rating categories (please see p8 of the previous version). However, we understand that the origin of the category descriptions might not have been clear. In the revised manuscript, we have clarified that these descriptions were taken verbatim from the Ramsøy and Overgaard paper as follows (p9 line 189): *These explanations were taken verbatim from Ramsøy and Overgaard (2004).*

1-5 *p13 §2: “...they were not VISIBLE even in the supraliminal group.”
Figures are numbered incorrectly in the text.*

Reply: We revised these points in the revised manuscript. We would like to thank Reviewer 1.

1-8 *Why does Fig. 3 not show the d' values as Fig. 2 does? The figure should also indicate the scale range [1..7] for the truth scores.*

Reply: As shown in the protocol, the recognition test was performed in the preliminary experiment, not the main experiment. Thus, Fig. 3, which shows the results of the main experiment, does not show the d' value. On the other hand, we revised the scale range for the truth scores in the revised manuscript.

Once again, we would like to express our gratitude to the editor and reviewers for their thoughtful and constructive comments. We hope that our revised manuscript is now suitable for publication in *Royal Society Open Science*.

Sincerely,

Kyoshiro Sasaki, Ph. D.

Associate Professor

Faculty of Informatics, Kansai University,

2-1-1, Ryozenji-cho, Takatsuki, Osaka, 569-1095, Japan.

E-mail: kyoshiro0920@gmail.com

TEL & FAX: +81-72-690-2447